# MicroRNAs Associated with Androgen Receptor and Metastasis in Triple-Negative Breast Cancer

**DOI:** 10.3390/cancers16030665

**Published:** 2024-02-04

**Authors:** Mamoun Ahram, Bayan Abu Alragheb, Hassan Abushukair, Randa Bawadi, Maysa Al-Hussaini

**Affiliations:** 1Department of Physiology and Biochemistry, School of Medicine, The University of Jordan, Amman 11942, Jordan; bawadiranda@hotmail.com; 2School of Medicine, The University of Jordan, Amman 11942, Jordan; bya0181543@ju.edu.jo; 3School of Medicine, Jordan University of Science and Technology, Irbid 22110, Jordan; hmabushukair182@med.just.edu.jo; 4Department of Pathology and Laboratory Medicine, King Hussein Cancer Center, Amman 11941, Jordan; mhussaini@khcc.jo

**Keywords:** breast cancer, TNBC, MicroRNA, androgen receptor, metastasis

## Abstract

**Simple Summary:**

Triple-negative breast cancer (TNBC) is notoriously aggressive and challenging to treat. Via their intracellular androgen receptor (AR), androgens play important roles in the biology of TNBC. MicroRNA (miRNA) molecules have been shown to mediate the biological actions of androgens. They also represent promising biomarkers for the diagnosis, prognosis, and therapy of TNBC. PCR arrays were utilized to profile the expression of 84 mRNA in human TNBC tissue samples classified according to AR expression and their metastasis status. Results revealed the association of only one miRNA, namely hsa-miR-205-5p, with metastasis. The levels of specific miRNAs such as hsa-miR-328-3p and hsa-miR-489-3p were higher in AR-positive than AR-negative TNBCs. Whereas the former was specifically higher in the metastatic AR-positive TNBC, hsa-miR-489-3p was more expressed in the non-metastatic subtype. These miRNAs may explain some of the biological effects of androgens on TNBC. Their use in the clinical setting needs further study.

**Abstract:**

It is crucial to identify novel molecular biomarkers and therapeutic targets for triple-negative breast cancer (TNBC). The androgen receptor (AR) is a regulator of TNBC, acting partially via microRNA molecules (miRNAs). In this study, we used PCR arrays to profile the expression of 84 miRNAs in 24 TNBC tissue samples, which were equally classified according to AR expression and/or metastasis. Several bioinformatics tools were then utilized to determine the potentially affected protein targets and signaling pathways. Seven miRNAs were found to be significantly more highly expressed in association with AR expression, including miR-328-3p and miR-489-3p. Increased expression of miR-205-3p was found to be significantly associated with metastasis. Certain miRNAs were specifically found to be differentially expressed in either metastatic or non-metastatic AR-positive tumors. A gene ontology (GO) analysis indicated biological roles in the regulation of transcription, cellular response to DNA damage, and the transforming growth factor-beta (TGF-beta) signaling pathway. The GO analysis also showed enrichment in kinase and transcription factor activities. The TGF-beta and a number of kinase-dependent pathways were also retrieved using the Kyoto Encyclopedia of Genes and Genomes (KEGG) enrichment analyses. This study offers an understanding of the role of AR in TNBC and further implicates miRNAs in mediating the effects of AR on TNBC.

## 1. Introduction

MicroRNA molecules (miRNAs) are short non-coding RNAs that regulate the levels of proteins via either blocking translation or driving mRNA degradation. Abnormal levels of miRNAs may result in the dysregulation of cell function, leading to pathological conditions such as cancer and cardiovascular and neurodegenerative diseases [1,2,3,4,5]. In relation to cancer, the affected proteins may be tumor drivers or suppressors, and, hence, miRNAs can play a role in tumorigenesis [6]. Some have considered miRNAs as “fine tuners” of cell behavior [7]. miRNAs may modulate tumor response to therapy [5], and they have recently been shown to be promising therapeutic tools [8,9,10].

Numerous research studies have investigated the differential expression of miRNAs in association with the molecular classes of breast cancer. These studies have shown that each cancer class has its unique miRNA profile, which is linked to a distinct biological behavior, therapeutic efficacy, and clinical outcome [11,12,13,14,15]. A more recent study by Kurozumi et al. reported that the dysregulation of specific miRNAs was associated with BRCA mutations, the immune system, epithelial–mesenchymal transition, cancer stem cell properties, and androgen receptor (AR) expression [16]. 

Triple-negative breast cancer (TNBC) is a notorious type of cancer that is difficult to treat and is associated with poor prognosis [17,18]. This class of breast cancer has been found to be heterogeneous and can be subdivided into different subtypes [19]. Interestingly, a considerable proportion (15–55%) of TNBC expresses the AR, suggesting that it may play a role in the biology of TNBC [20,21]. Indeed, AR-positive TNBCs have a worse chemotherapeutic sensitivity and lower complete remission rates following neoadjuvant treatment [21] and are associated with increased metastasis [22,23]. The effects of AR are not limited to a subtype of TNBC that is characterized by a high expression of the AR, hence termed luminal AR; it extends to other subtypes of TNBC [24]. Therefore, AR may be an intriguing biomarker and therapeutic target. For example, anti-androgen treatment with enzalutamide, an AR antagonist, has modest clinical efficacy in phase II clinical study [25]. In addition, several preclinical and early clinical investigations have demonstrated that AR can be a targeted therapy in AR-positive TNBC [26]. 

The differential expression of miRNAs in TNBC has also been reported [27]. An earlier study has shown that miR-363 is induced in AR-expressing breast MDA-MB-453 cancer cells and that this miRNA targets the mRNA; hence, the expression of IQ motif and WD repeats-1 (IQWD1) [28]. We revealed multiple androgen-mediated changes in the levels of miRNAs in the MDA-MB-453 cells [29]. We also reported that treatment of TNBC cells with dihydrotestosterone induced the expression of miR-328-3p, which targeted the mRNA of the cell adhesion molecule CD44, reducing protein expression and altering cell adhesion and migration [30]. 

Due to the functional and clinical significance of both AR and miRNAs in TNBC, and the lack of biomarkers for this disease, it is necessary to reveal the role of androgens and AR in regulating the expression of miRNAs and controlling the behavior of TNBC, in order to identify novel diagnostic and/or therapeutic targets. Therefore, we aimed to elucidate the levels of a number of breast cancer-specific miRNAs in association with AR expression and metastasis status. For this purpose, human TNBC tissue samples were utilized, and the levels of miRNAs were assessed via PCR arrays.

## 2. Materials and Methods

### 2.1. Patient Selection

Patients with a confirmed diagnosis of TNBC were included in our study with available resected tissue for miRNA profiling. Extracted data included clinical variables (age, cancer laterality, overall survival defined from diagnosis) and pathological variables (tumor size, AJCC TNM stage, grade, histology, and lymphovascular invasion). Ethical approval for the study was obtained from the institutional review boards of Jordan University Hospital (#10/2015/4953) and King Hussein Cancer Center (#17KHCC46 on 14 March 2018). 

### 2.2. AR Detection by Immunohistochemistry (IHC)

Twenty-four formalin-fixed, paraffin-embedded TNBC tissue samples were retrieved from the Department of Pathology and Laboratory Medicine, King Hussein Cancer Center (Amman, Jordan). Hematoxylin and eosin staining was performed using standard procedures. The tissue samples were evaluated for the expression of AR by immunohistochemistry and were classified accordingly as AR-positive or -negative, as well as according to their metastasis status. For the immunohistochemistry (IHC), 5 µm thick tissue sections were cut and placed on coated slides with tissue sections of a known positive control placed near the non-frosted edge of the slide. The mouse polyclonal anti-AR antibody (MS-433-R7; Thermo Medical Co., Somerset, NJ, USA) was used for AR staining, which was carried out using Ventana Benchmark Ultrastainer (Ventana Medical Systems, Oro Valley, AZ, USA). The slides were examined by a pathologist who was totally blinded to the reported clinical data. The examined sections were reported as AR-positive when more than 1% of tumor cells had positive nuclear immunostaining. External AR-negative cases were used as a negative control to ensure there were no technical issues.

### 2.3. Preparation and Relative Quantification of miRNA in Tissue Samples

For expression profiling, each tissue sample represented a unique case and comprised one 10 μm tissue section placed in a small microcentrifuge tube. All reagents for the PCR arrays were obtained from Qiagen Inc. (Valencia, CA, USA). A sample analysis for microRNA levels in the tissue samples was performed as previously described [31]. Briefly, total RNA was extracted from the tissue sections using the RNeasy FFPE Kit according to the standard protocol of the manufacturer (Qiagen, Hilden, Germany). The concentration and purity of total RNA isolates were spectrophotometrically measured using the NanoDrop 2000 (Thermo Fisher Scientific, Waltham, MA, USA). Complementary DNA (cDNA) was synthesized using the miScript II RT Kit (Qiagen). The expression of a panel of 84 cancer-related miRNAs using miScript miRNA Human Breast Cancer PCR Array (MIHS-109Z) was determined, where different control snoRNAs/snRNAs (SNORD61, SNORD68, SNORD72, SNORD96A, and RNU6B/RNU6-2) were used as housekeeping genes. A PCR was performed in Bio-Rad’s IQ Real-Time System (Hercules, CA, USA). 

Relative quantification for the miRNAs was performed based on the ∆∆C_t_ method using the RT^2^ Profiler PCR Array Data Analysis web portal provided at Qiagen’s web portal. The samples were analyzed for miRNA expression according to the following eight criteria with the former assigned as the control group: (1) AR-positive versus AR-negative, (2) non-metastatic versus metastatic, (3) AR-positive and metastatic versus AR-negative and metastatic, (4) AR-positive and non-metastatic versus AR-negative and non-metastatic, (5) AR-negative and metastatic versus AR-negative and non-metastatic, (6) AR-positive and metastatic versus AR-negative and non-metastatic, (7) AR-positive and metastatic versus AR-positive and non-metastatic, and (8) AR-negative and metastatic versus AR-positive and non-metastatic. 

### 2.4. Bioinformatics Analyses

To further validate the associations between AR and miRNA expression, datasets from the Cancer Genomic Atlas Breast Invasive Carcinoma cohort (TCGA-BRCA, n = 1084) and the Gene Expression Omnibus (accession: GSE19536, n = 101) were utilized. The analyses were limited to TNBC patients in both datasets (TCGA-BRCA, n = 171, GSE19536, n = 15). Calculations of Spearman’s Rho correlation coefficients were used to correlate the mRNA expression of each miRNA and AR for both, and Pearson R2 correlation coefficients were used for the TCGA. A heatmap was also generated to illustrate the linkage of all miRNAs to AR expression in all TNBC patients. 

Potential target genes of differentially expressed (DE) miRNAs were identified for each miRNA using miRWalk2.0. Selection criteria for the target genes included those that were validated and found to correlate with the miRNA in TargetScan or miRDB databases. A gene ontology (GO) analysis (http://geneontology.org/, accessed on 29 January 2024) was performed at three levels: molecular function (MF), biological process (BP), and cellular component (CC). A pathway analysis was performed using the Kyoto Encyclopedia of Genes and Genomes database (KEGG) (www.genome.jp/kegg, accessed on 29 January 2024). Functional enrichment and pathway enrichment analyses were performed by using the Database for Annotation, Visualization and Integrated Discovery (DAVID) web tool (http://david-d.ncifcrf.gov, accessed on 29 January 2024). Both GO and KEGG analyses were conducted for the target genes using the DAVID 6.8 bioinformatics tool (*p*-value < 0.05). The OncomiR online database (http://www.oncomir.org/, accessed on 29 January 2024) was utilized to correlate selected miRNAs with clinicopathologic criteria, including metastasis, lymph node infiltration, and tumor size [32]. 

### 2.5. Statistical Analyses

Descriptive measures included mean and standard deviation (SD) for continuous data, while counts and percentages were used for categorical data. The log-rank test was used to compare survival between high- and low-expression groups, and Kaplan–Meier plots were used to visualize the overall survival probability among patients. Comparison between the expression levels for miRNAs of interest across clinicopathological variables was carried out using the independent Student’s *t*-test in case of data normality; otherwise, the Mann–Whitney U test was used. The analysis of variance (ANOVA) test was used to compare miRNA expression values across variables with more than two groups. The statistical analysis was performed using JAMOVI software, version 2.3, (https://www.jamovi.org, accessed on 29 January 2024), and significance was set at *p* < 0.05. 

## 3. Results

### 3.1. Included Patients

A total of 24 patients were included and underwent miRNA profiling, 16 of whom had complete clinicopathological data. The mean age was 52.4 (SD: 11.1). The majority of patients had right-sided (68.85) breast cancer and were Jordanians (81.3%). Half of the cases had metastases, yet 61.5% were lymph node-negative. More than half (57.1%) had a tumor size status of 2, and 68.8% displayed lymphovascular invasion. Three-fourths of the patients were still alive, and the mean survival time was 58.2 months. A complete description of patients’ characteristics is shown in Table 1. 

### 3.2. Identification of DE miRNAs

The samples were classified according to their expression status of the AR or metastasis. Accordingly, comparative miRNA expression analyses of eight different combinations, described in the methodology section, were carried out. The differentially expressed miRNAs per category are listed in Table 2, and the original raw data of the threshold cycle per miRNA for each sample are presented in Appendix A. Seven miRNAs were found to be associated with AR expression in TNBC. Three of them (miR-17-5p, miR-20a-5p, and miR-20b-5p) were down-regulated in AR-positive TNBC and 4 miRNAs (miR-193b-5p, miR-328-3p, miR-485-5p, miR-489-3p) were up-regulated. miR-205-5p was the only miRNA whose expression was altered when the samples were compared for the metastasis status where it was found to be up-regulated metastatic TNBC tissues. 

Upon examining the differentially expressed miRNAs according to both AR expression and metastasis, some of the aforementioned miRNAs persisted, others did not, and new ones appeared. For example, miR-328-3p and miR-193b-3p were found to be highly expressed in AR-positive metastatic cases. A new miRNA, miR-181b-5p, appeared to be up-regulated in the former tumor type. In AR-positive non-metastatic tumors, the expressions of miR-223-3p, miR-26a-5p, miR-26b-5p, and miR-489-3p were up-regulated. The expression of miR-205-5p was also found to be up-regulated in metastatic tumors but was not different within AR-positive cases. Four miRNAs (miR-193b-5p, miR-223-3p, miR20a-5p, and miR-328-3p) were up-regulated in AR-positive metastatic tumors versus AR-negative non-metastatic tumors, and one miRNA (miR-489-3p) was up-regulated in AR-positive non-metastatic tumors versus AR-negative metastatic tumors.

To further validate the association between AR expression and the 11 miRNAs whose expression was associated with AR, we utilized two external gene expression databases in TNBCs, TCGA-BRCA and GSE19536, which included 171 and 15 TNBC patients, respectively. The TCGA dataset included only two miRNAs, miR-17-5p and miR-193b-5p. Based on Pearson’s correlation, there was a considerable association between the expression of both AR and miR-17-5p (R2 = 0.18) with a significant *p*-value that equaled 0.0193. According to Spearman’s correlation, the inverse association of expression was reasonable Rho = −0.12), although it was not significant (*p*-value = 0.131). The inverse expression association between AR and miR-193b-5p, however, was weak and insignificant. The data can be found in the Appendix A.

Strong agreements were found between the correlation of the expression of AR and some of the 11 AR-associated miRNAs generated by the PCR arrays in this study and the GEO dataset (Appendix A). However, the low sample number (n-15) probably hindered these agreements from reaching statistical significance. These correlations were apparent for miR-328-3p (Rho = 0.421) and miR-489-3p (Rho = 0.296), and miR-223-3p (Rho = 0.271), miR-26a-5p (Rho = 0.261), miR-26b-5p, miR-181d-5p (Rho = −0.368), and miR-17-5p (Rho = −0.121). Interestingly, a heatmap generated based on the GEO dataset clustered miR-328-3p, miR-489-3p, and miR-485-5p in AR-positive TNBC cases and miR-17-5p, miR-20a-5p, miR-20b-5p, and miR-181d-5p in AR-negative tumors.

### 3.3. Identification of Target Genes and Enrichment Analysis

Predicted target genes for the up-regulated and down-regulated DE-miRNAs that were identified using miRWalk2.0 are presented in Appendix A. DAVID 6.8 was utilized for the enrichment analysis of GO and KEGG pathways. GO includes three main categories: biological process (BP), cellular component (CC), and molecular function (MF). The significantly associated and relevant GO and KEGG pathways are listed in Figure 1. In the biological process (BP) term, the predicted target genes were mainly associated with cellular response to DNA damage and the regulation of transcription (Figure 1A). In the cellular component (CC) term, the predicted targeted genes were enriched in the centrosome, nucleus, and transcription factor complex (Figure 1B). In the molecular function (MF) term, the predicted targeted genes were mainly related to signaling pathways related to kinase activities (Figure 1C). The analysis of the KEGG pathway revealed that the predicted targeted genes were mainly involved in TGF-beta signaling, cellular senescence, autophagy, endocrine resistance, and several kinase-dependent pathways (Figure 1D).

### 3.4. Clinicopathological Associations 

Patients with lymph node involvement (N2 and N3) had significantly higher expression of miR-205-5p compared to N1 and N0 patients. In addition, distant metastasis was associated with increased expression of miR-205-5p and miR-485-5p (Figure 2). Based on tumor location, miR-205-5p was significantly expressed at a higher level on the left side (mean: 24.4 vs. 22.3, *p* = 0.046). Regarding grade, miR-17-5p (*p* = 0.036) and miR-20a-5p (*p* = 0.039) had higher levels in low-grade patients. miR-17-5p, miR-193b-3p, miR-20a-5p, miR-26a-5p, miR-26b-5p, and miR-485-5p all had higher expression in patients with lymphovascular invasion (*p* < 0.05). 

To further test the prognostic value of the AR-associated miRNAs, we used the OncomiR online database. As shown in Table 3, dysregulation of the expression of several of the miRNAs was associated with the metastatic status and lymph node infiltration, including miR-17-5p, miR-193b-3p, and miR-205-5p.

### 3.5. Survival Analysis

Included patients with survival data (n = 16) were categorized into high and low expressions using the median expression as a cut-off point. There were no significant differences between expression levels for all eleven identified miRNAs in association with patent survival (Appendix A), although patients with a low miR-193b-3p tended to be associated with increased overall survival (Figure 3). 

## 4. Discussion

The role of miRNAs in modulating breast cancer is well documented, and the potential use of these molecules in diagnosing and treating TNBC is believed to be promising. This study has revealed several miRNAs that are associated with AR expression and/or metastasis. Prominent AR-regulated miRNAs included miR-328-3p, miR-489-3p, and miR-193b-3p. Some other miRNAs appeared to be AR-regulated, but only in certain conditions depending on the metastatic status of the tumors. These included miR-26a-5p, miR-26b-5p, miR-223-3p, and miR-181d-5p. Moreover, metastasis appeared to be associated with miR-205-5p, particularly in AR-negative tumors. The association of these miRNA molecules with breast cancer will be discussed later in light of previous investigations. Although some are in line with these reports, others such as miR-205-5p may be conflicting.

miR-205 is one of the most extensively studied miRNAs in cancer. It is a well-known suppressor of cancer metastasis, and its low expression is associated with increased metastasis. A recent study has shown that significantly higher levels of miR-205 were found in the ER-positive compared with the ER-negative tumors, and that patients with a low-miR-205 expression level had significantly higher 5-year survival rates [34]. This miRNA has been linked to blocking epithelial-mesenchymal transition [34,35] and chemoresistance [36,37]. These results are consistent with our finding that a lower expression of miR-205 is associated with increased distant metastases. However, it is important to note that a higher expression of miR-205-3p was found in metastatic tumors versus non-metastatic tumors and, particularly, in the AR-negative cohorts.

miR-489-3p has been found to increase the sensitivity of breast cancer cells via targeting cyclin-dependent kinase 1 (circCDK1) [38]. However, the effect of this miRNA in cancer depends on the cancer type. For example, whereas miR-489-5p was found to promote the metastasis of osteosarcoma [39] and non-small cell lung cancer [40], it suppresses the proliferation of bladder cancer [41], glioblastoma [42], and melanoma [43]. This miRNA is under-regulated by competing endogenous noncoding RNA molecules [44,45,46,47].

Another intriguing miRNA is miR-485-5p, which was also found to be lower in tumors with distant metastasis, although it was found to possess a higher expression in patients who did not have a lymphovascular invasion. This miRNA acts as a suppressor of various tumor types, including breast cancer [48,49,50,51], and is also under regulation by upstream competitive endogenous circular and long noncoding RNA molecules [52,53,54,55].

Previously, we have reported the upregulation of miR-328-3p in a TNBC cell line that expresses small quantities of AR upon activation of the receptor via treating them with dihydrotestosterone [33]. We then found that this miR-328-3p targets the CD44 receptor, influencing their migration and adhesion properties [30]. Interestingly, miR-328-3p was found to be down-regulated, particularly in metastatic breast cancer [56]. Contrary to the latter study, not only is miR-328-3p associated with AR expression as shown in this study, and is regulated by AR as reported earlier, but this study has found that it is more expressed in metastatic TNBC. The same miRNA could also target the mRNA of the drug transporter ABCG2, reducing its protein level [57]. However, the chemoresponse was not altered as a result of this targeting, enforcing the role of miRNAs as “fine-tuners” of cell function [58]. 

Treatment of the TNBC cell line MDA-MB-231 with dihydrotestosterone augmented the expression of two other miRNA molecules of the same family, miR-26a-5p and miR-26b-5p [33]. In line with these results, both miRNAs appeared to be regulated by the AR where higher levels were found in AR-positive tumors relative to the AR-negative ones, but only in the non-metastatic type. Both molecules appear to have suppressive effects on breast cancer. For example, both inhibit breast cancer cell proliferation, with miR-26a-5p also being associated with better patient survival [59]. They also increase the chemosensitivity of breast cancer cells, including MDA-MB-231 [60,61]. It is noteworthy that we also found that miR-26b-5p promotes chemosensitivity in a cell-specific manner (Alsawalha and Ahram, unpublished data). 

Upon predicting the pathways that are influenced by the differentially expressed miRNA molecules, it can be noted that pathways are related to transcriptional activity as would be expected for AR. Multiple intriguing pathways appear to be controlled by miR-17-5p, miR-20a-5p, and miR-20b-5p, which are part of a cluster termed the miR-17-92a cluster. The three miRNAs that belong to that cluster were found to be considerably down-regulated in AR-positive TNBC. This cluster has been reported to be regulated by AR in prostate cancer in association with autophagy [62]. These miRNAs are known to be pro-proliferative and are strongly associated with metastasis and poor patient survival [63]. Upregulation of the cluster’s components can differentiate two TNBC subclasses, basal-like 1 and 2 [64], highlighting the possible usefulness of miRNAs. Importantly, the fact that AR expression is associated with decreased expression of these miRNAs suggests that it is a good prognostic marker.

A strong functional association between AR and TGF-beta has also been reported whereby both factors positively regulate each other in a feedback loop in TNBC [65]. Interestingly, the miR-17-92a cluster has been predicted to target the TGF-beta signaling pathway [66,67]. Experimental validation is needed to confirm the AR-TGF-beta-miR-17/20a axis. TGF-beta is an inducer of EMT just like AR.

The role of androgens in breast cancer requires further thorough investigations. We have shown that the treatment of breast cancer cells with dihydrotestosterone elevated the levels of miR-328-3p in TNBC [30,33] This study further confirmed that this miRNA is androgen-regulated. However, such upregulation may not necessarily influence the molecular pathways that the mRNA targets of this miRNA are involved in. For example, whereas upregulation of miR-328-3p reduced its target CD44 and, hence, the ability of the cells to adhere to hyaluronic acid and motility, the same was not true for another target, that is ABCG2, whose downregulation did not result in chemosensitivity [57]. Rather, it resulted in androgen-stimulated chemoresistance via the transcriptional regulation of apoptotic regulatory genes [68]. These results suggest that transcriptional control may overcome miRNA-mediated regulation of protein levels and indicate that miRNAs are “fine-tuners” of cell function, as proposed earlier [7]. Further, they may pinpoint the complexity of molecular networks. 

## 5. Conclusions

Given the plethora of downstream effects of miRNA molecules and the biological roles of AR in TNBC, their functional association may aid in understanding the etiology and progression of TNBC. In addition, the influence of both AR and miRNAs on tumor response to treatment suggests that they can be utilized to facilitate the eradication of cancer cells. It is of note that the common pathways that have been hypothesized to be regulated by the identified miRNAs are also AR-regulated. Some of these pathways are relevant in altering the response of TNBC to therapy by increasing cell sensitivity or reducing resistance [69]. The notion of miRNAs as biomarkers of TNBC [70,71,72] is still in its infancy. The heterogeneity of the disease, the incomplete annotation of miRNAs, and the variable technologies used for the identification of miRNAs are clear obstacles. In addition, TNBC constitutes a small proportion (~10%) of breast cancer, and there is a need to analyze a larger number of samples to reach definitive conclusions. The latter can be considered a limitation of the study herein. Further studies are needed and a larger number of samples is warranted. Nevertheless, the findings of this study suggest the involvement of miRNAs in mediating the effects of androgens on the behavior of breast cancer, and a step towards better understanding the disease.

## Figures and Tables

**Figure 1 cancers-16-00665-f001:**
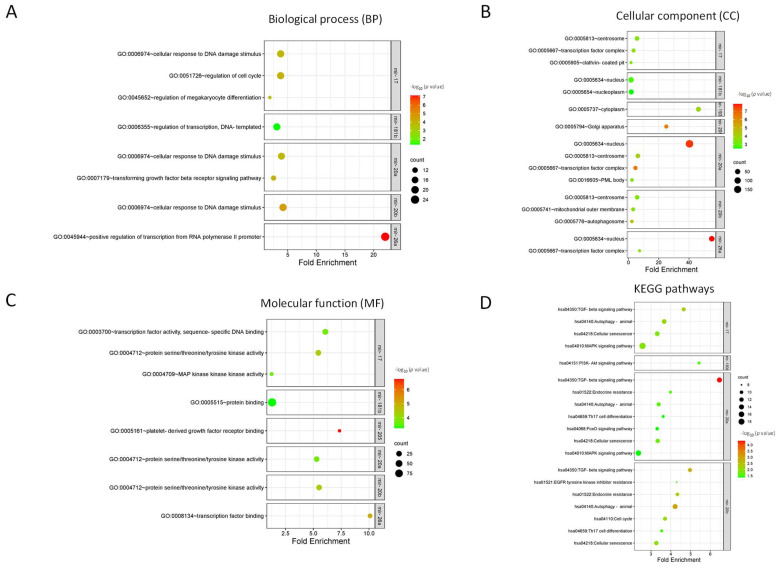
Gene ontology enrichment for biological processes (**A**), cellular components (**B**), molecular functions (**C**), and KEGG pathway enrichment (**D**) of predicted targeted genes.

**Figure 2 cancers-16-00665-f002:**
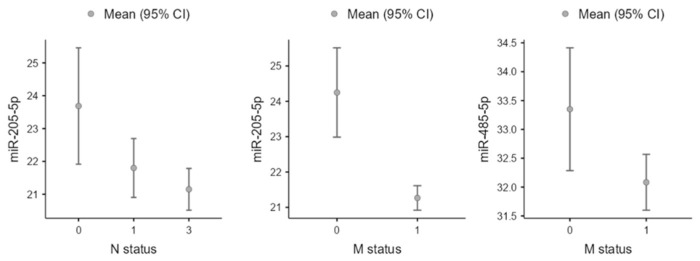
Comparison between hsa-mir-205-5p and hsa-mir-485-5p expression across metastasis (M) and lymph node (N) statuses (*p* < 0.05).

**Figure 3 cancers-16-00665-f003:**
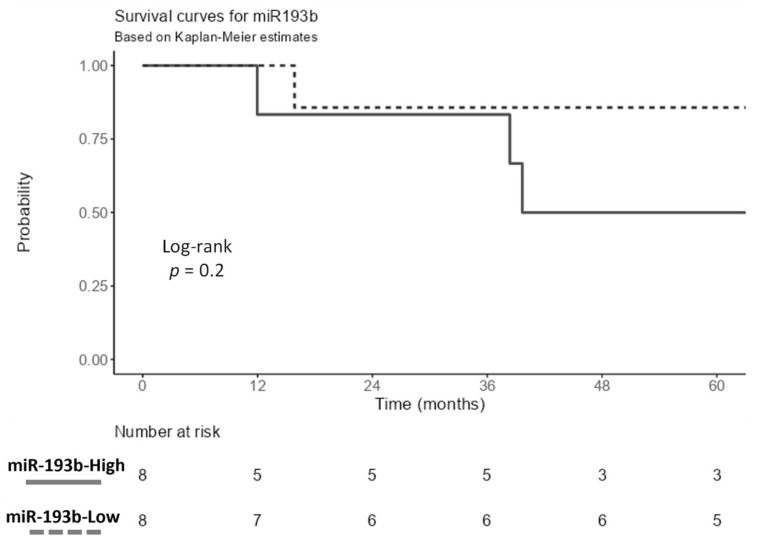
Kaplan–Meier plot demonstrates overall survival probability based on hss-mir-193b-3p expression.

**Table 1 cancers-16-00665-t001:** Clinicopathological characteristics of patients (n = 16 ^a^).

Criteria	Mean (SD)/Count (%)
Age at diagnosis (years)	52.4 (11.1)
Laterality	
Right	11 (68.8)
Left	5 (31.3)
Nationality	
Jordanian	13 (81.3)
Others	3 (18.7)
Histology	
Not otherwise specified	14 (88.5)
Apocrine	2 (12.5)
Grade	
High (3)	14 (87.5)
Low (1 and 2)	2 (12.5)
Lymphovascular invasion	
Present	11 (68.8)
Absent	5 (31.3)
T status	
1	3 (21.4)
2	8 (57.1)
3	3 (21.4)
N status	
0	8 (61.5)
1	3 (23.1)
2	0 (0)
3	2 (15.4)
M status	
0	12 (50)
1	12 (50)
Vital status: Alive	12 (75)
Median overall survival (months)	58.2

^a^ n = 16 for all categories except for the presence of metastasis where the data were available for all 24 samples.

**Table 2 cancers-16-00665-t002:** Differentially expressed miRNA molecules in TNBC tissues according to AR and/or metastasis statuses.

miRNA	Expression Fold	*p*-Value	Notes
AR-positive versus AR-negative			
miR-17-5p	−3.6	0.031	Down-regulated
miR-193b-3p	2.6	0.007	Up-regulated
miR-20a-5p	−3.93	0.012	Down-regulated
miR-20b-5p	−5.12	0.028	Down-regulated
**miR-328-3p ^#^**	3.11	0.031	Up-regulated
miR-485-5p	1.73	0.027	Up-regulated
**miR-489-3p**	3.29	0.009	Up-regulated
2. Metastatic versus non-metastatic			
miR-205-5p	4.41	0.029	Up-regulated
3. AR-positive, non-metastatic versus AR-negative, non-metastatic			
miR-223-3p	2.33	0.04	Up-regulated
**miR-26a-5p**	2.1	0.04	Up-regulated
**miR-26b-5p**	2.77	0.03	Up-regulated
**miR-489-3p**	3.11	0.01	Up-regulated
4. AR-positive, metastatic versus AR-negative, metastatic			
miR-181d-5p	2.52	0.04	Up-regulated
miR-193b-3p	3.21	0.03	Up-regulated
**miR-328-3p**	2.61	0.03	Up-regulated
5. AR-negative, metastatic versus AR-negative, non-metastatic			
miR-205-5p	9.08	<0.01	Up-regulated
6. AR-positive, metastatic versus AR-positive, non-metastatic			
None			
7. AR-positive, metastatic versus AR-negative, non-metastatic			
miR-193b-3p	2.8	0.04	Up-regulated
miR-205-5p	5.86	0.01	Up-regulated
miR-20b-5p	−7.82	0.03	Down-regulated
**miR-328-3p**	2.38	0.04	Up-regulated
8. AR-positive, non-metastatic versus AR-negative, metastatic			
**miR-489-3p**	2.67	0.04	Down-regulated

**^#^** miRNAs in bold indicate that the miRNA was found to be AR-regulated in the TNBC MDA-MB-231 cells in a previous study [33].

**Table 3 cancers-16-00665-t003:** Association of selected miRNAs with the clinicopathologic features according to the Oncomir database.

miRNA Name	Clinical Parameter	ANOVA *p*-Value	ANOVA FDR ^#^	Multivariate Log Rank *p*-Value	Multivariate Log Rank FDR
miR-17-5p	Pathologic N Status	**4.43 × 10^−4^**	**2.58 × 10^−2^**	3.07 × 10^−1^	9.11 × 10^−1^
	Pathologic Stage	**2.28 × 10^−2^**	2.24 × 10^−1^	4.59 × 10^−1^	9.98 × 10^−1^
	Pathologic T Status	**3.70 × 10^−4^**	**5.48 × 10^−3^**	4.70 × 10^−1^	9.98 × 10^−1^
miR-193b-3p	Pathologic M Status	**2.25 × 10^−3^**	**1.50 × 10^−2^**	3.19 × 10^−1^	9.48 × 10^−1^
	Pathologic T Status	**1.47 × 10^−2^**	8.85 × 10^−2^	1.43 × 10^−1^	7.29 × 10^−1^
miR-20a-5p	Pathologic M Status	**4.88 × 10^−2^**	1.70 × 10^−1^	8.44 × 10^−1^	9.97 × 10^−1^
	Pathologic N Status	**4.82 × 10^−3^**	7.67 × 10^−2^	8.35 × 10^−1^	9.98 × 10^−1^
	Pathologic T Status	**1.08 × 10^−3^**	**1.25 × 10^−2^**	9.84 × 10^−1^	9.98 × 10^−1^
miR-205-5p	Pathologic M Status	**1.80 × 10^−3^**	**1.27 × 10^−2^**	1.41 × 10^−1^	7.17 × 10^−1^
	Pathologic N Status	**1.15 × 10^−2^**	1.04 × 10^−1^	1.14 × 10^−1^	6.66 × 10^−1^
	Pathologic Stage	**5.75 × 10^−4^**	**2.41 × 10^−2^**	1.15 × 10^−1^	6.88 × 10^−1^
	Pathologic T Status	**1.26 × 10^−6^**	**1.13 × 10^−04^**	1.04 × 10^−1^	6.51 × 10^−1^
miR-223-3p	Pathologic N Status	**4.71 × 10^−2^**	2.19 × 10^−1^	1.49 × 10^−1^	7.31 × 10^−1^
	Pathologic Stage	**4.56 × 10^−2^**	3.13 × 10^−1^	1.64 × 10^−1^	7.75 × 10^−1^
	Pathologic T Status	**6.68 × 10^−3^**	5.01 × 10^−2^	1.40 × 10^−1^	7.29 × 10^−1^
miR-26a-5p	Pathologic M Status	8.70 × 10^−1^	9.41 × 10^−1^	**4.75 × 10^−2^**	5.27 × 10^−1^
	Pathologic N Status	4.80 × 10^−1^	7.91 × 10^−1^	**2.50 × 10^−2^**	4.12 × 10^−1^
	Pathologic Stage	9.15 × 10^−2^	4.37 × 10^−1^	**1.81 × 10^−2^**	4.02 × 10^−1^
	Pathologic T Status	**4.51 × 10^−4^**	**6.30 × 10^−3^**	**2.75 × 10^−2^**	3.96 × 10^−1^
miR-26b-5p	Pathologic M Status	**9.24 × 10^−3^**	5.08 × 10^−2^	1.33 × 10^−1^	7.11 × 10^−1^
miR-328-3p	Pathologic M Status	**1.89 × 10^−2^**	8.63 × 10^−2^	1.14 × 10^−1^	6.84 × 10^−1^
	Pathologic N Status	**1.25 × 10^−2^**	1.10 × 10^−1^	9.13 × 10^−2^	6.27 × 10^−1^
	Pathologic T Status	**2.11 × 10^−2^**	1.11 × 10^−1^	7.64 × 10^−2^	6.13 × 10^−1^
miR-489-3p	Pathologic M Status	**2.02 × 10^−2^**	9.06 × 10^−2^	8.49 × 10^−1^	9.97 × 10^−1^

^#^ FDR, false discovery rate. *p*-values in bold are significant (<0.05).

## Data Availability

The authors confirm that the data supporting the findings of this study are available within the article and/or its Appendix A.

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
