# Peer review of "MicroRNAs Associated with Androgen Receptor and Metastasis in Triple-Negative Breast Cancer"

_cancers, 2024, doi:10.3390/cancers16030665_

Round 1
Reviewer 1 Report
Comments and Suggestions for Authors
cancers-2793622 Review
MicroRNAs Associated with Androgen Receptor and Metastasis in Triple-Negative Breast Cancer
This article is written well, but lacks in new knowledge and grounds. Therefore, I require Major Revision.
Major point
・Please clarify the purpose of this study in “Introduction” section.
・Please describe the analysis method in this research in more detail.
・Additional experiments are necessary to prove this theme. This study needs to increase the number of cases to increase reliability (Future clinical application is expected).
・Please show results a little more clearly.
・Please indicate future clinical applications in the "Discussion section".
Minor point
・The sentence of this paper has many careful mention errors. Please review it.
Author Response
Major comments:
Comment 1: Please clarify the purpose of this study in the “Introduction” section.
Response: We thank the reviewer for highlighting this. We have modified the last paragraph in the Introduction section to emphasize the purpose of the study (page 6, lines 5-11).
Comment 2: Please describe the analysis method in this research in more detail.
Response: We agree with the reviewer that additional information can aid other investigators in understanding and repeating the study. We have therefore added another subsection on page 7, line 7 entitled “AR detection by immunohistochemistry (IHC)”. We have also modified and elaborated on the “Bioinformatics analysis” subsection (page 9, line 9)
Comment 3: Additional experiments are necessary to prove this theme. This study needs to increase the number of cases to increase reliability (Future clinical application is expected).
Response: We completely agree with the reviewer that additional samples are needed to increase the reliability of the results. However, it is normally difficult to collect 24 tissue samples given that TNBC samples represent around 10% of breast cancer cases and the fact that we divided the cases into four categories according to both metastasis and the expression of AR statuses. The number of samples per category was sufficient to reach a meaningul conclusion and generate reliable statistical significance. In addition, the fact that our results resemble what we have generated previously upon treating TNBC cell line with dihydrotestosterone strengthens the results presented in the study. Yet, we have mentioned in the Conclusion section (page 26, lines 3-6) the need to evaluate a larger number of samples and that the relatively small number of samples was a limitation of the study.
Comment 4: Please show results a little more clearly.
Response: We could see that some of the figures were somewhat fuzzy, especially the text. We have therefore, modified the figures by writing the text on top of the orginal text exactly as it was written. The text appears sharper. In addition, as can be seen from the tracked changes, more details have been throughout the results section to provide more clarity to the reader.
Comment 5: Please indicate future clinical applications in the "Discussion section".
Response: We have discussed the use of miRNA molecules in the clinical setting for disease diagnosis, prognosis, and treatment throughout the Discussion section when describing the significance of each miRNA. In the Conclusion section, we described the potential use of miRNA as biomarkers or for treatment and have added additional references. (see page 25, line 14)
Minor point
Comment 6: The sentence of this paper has many careful mention errors. Please review it.
Response: We thank the reviewer for pointing this out. We have rechecked the language of the manuscript. We also used Grammarly for re-checking.

Reviewer 2 Report
Comments and Suggestions for Authors
The manuscript from Ahram et al. reported that the microRNA expression level is related with triple-negative breast cancer, especially, seven miRNAs such as miR-328-3p and miR-489-3p are highly expressed in association with AR expression. They also found that miR-205-3p was significantly associated with cancer metastasis. Their results strongly highlight the role of miRNAs in mediating the effects of androgens on the behavior of breast cancer. The paper provides a comprehensive analysis of microRNA abundance in cancer tissue and their findings would definitely benefit the triple-negative breast cancer field. The results from current manuscript will be well received by TNBC field, offering implications for cancer treatment. The current manuscript is well-written and organized. However, in order to enhance the manuscript, I believe there is one concern that the authors should address.
1. The main experiment they performed is Relative quantification for the miRNAs on TNBC tissue samples. However, they didn’t provide any original information or raw data about experiments in main text or supporting information.
2. The supporting information is empty, the author should address this issue.
Comments on the Quality of English LanguageLanguage is fine, no major problem is found.
Author Response
Comment: they didn’t provide any original information or raw data about experiments in main text or supporting information.
Comment: 2. The supporting information is empty, the author should address this issue.
Response: We agree with the reviewer that raw data should have been provided. We have now included the raw data with our submission.
